# Propagate deeper and Adaptive Graph Convolutional Networks

**Sisi Zhang**
School of Computer Science and Engineering
Northeastern University
Shenyang, China
shujizhangsisi@126.com

**Lun Du** *
Data Knowledge Intelligence Group
Microsoft
Beijing, China
lun.du@microsoft.com

**Fan Li**
School of Computer Science and Engineering
Northeastern University
Shenyang, China
lif119@foxmail.com

**Ge Yu**
School of Computer Science and Engineering
Northeastern University
Shenyang, China
yuge@mail.neu.edu.cn

**Mengyuan Chen**
Beijing Normal University
Beijing, China
chenmy.sharon@gmail.com

## ABSTRACT

Graph Convolutional Networks (GCNs) are the basic architecture for handling graph-structured data. How to deepen GCNs like Convolutional Neural Networks (CNN) and Transformer to improve their capability has always been a challenge. As the number of layers increases, the performance of GCNs degrades, which is commonly attributed to over-smoothing but is constantly debated. In this paper, we eliminate the equivalence between model degradation and over-smoothing or gradient vanishing and propose a systematic solution, an Adaptive DeepGCN (ADGCN) architecture, which makes the model the potential to address all issues. We locate learnable parameters in optimal locations to enable adaptive adjustments for various types of graph-structured data. We conduct experiments on real-world datasets to verify the stability and adaptability of our architecture.

## 1 INTRODUCTION

The studies have demonstrated that deeper architectures can increase the expression ability of deep neural networks in many domains, however, model degradation refers to the decline in generalization performance after stacking many GCN layers. Many studies have investigated the causes of it. We discover that there are causes of diversification that depend on different concerns. Some studies have focused primarily on over-smoothing caused by multiple graph convolution operations, others have concentrated on gradient vanishing, and still, others have expanded their perspective to consider the generalization gap based on the Lipschitz constraint to identify the key factors which can enhance model performance (Yang et al., 2020; Chen et al., 2020a; Feng et al., 2020; Zhu et al., 2021; Yan et al., 2021; Chien et al., 2021; Godwin et al., 2022; Cong et al., 2021; Zhang et al., 2022; Huang et al., 2023).

We argue that model degradation is a complex problem that is not caused by a single issue but rather by the coexistence of various factors. The differences in conclusions among previous studies can be attributed to the alteration of key factors revealed due to different settings. For instance, some studies focus on models with fewer than 10 layers, which exhibit normal gradients and low complexity

---

*Corresponding Author

but suffer from over-smoothing. As the number of layers increases, the problem of gradient vanishing becomes more pronounced, potentially overshadowing other issues. We propose a systematic solution, an Adaptive DeepGCN (ADGCN) architecture, which makes the model the potential to address all issues. It is beneficial to consistently manage the interaction between various components through end-to-end learning in neural networks. We place learnable parameters at the appropriate locations so that the model has space to make adaptive adjustments to different graph-structured data. The effect is comparable to the architecture of precise manual parameter modification.

The ADGCN architecture consists of three components that regulate the model's information propagation, control the model complexity, and introduce the attention mechanism as a control switch for selective aggregation of information across layers. Through experiments, we gain a deeper understanding of the necessity of the "systematic" approach. For instance, when considering only reducing the model's complexity (i.e., the number of parameters in the model is excessive for the current dataset), there is an improvement in the generalization ability and training effectiveness compared to the basic GCN model. However, the degradation still persists.

## 2 ADAPTIVE DEEPGCN

When the performance of GCNs combined with residual connections or other technologies is still reduced (Li et al., 2019), it is clear that only guaranteeing the training accuracy (roughly equal to 1) is not enough. Additionally, as the number of layers increased, the performance of the GCN variants, namely GCNII (Chen et al., 2020b) and Decoupled GCN (DGCN) (Cong et al., 2021), improved. In order to create a more stable and adaptable deep GCN architecture, we also propose the ADGCN.

The ADGCN model can be mathematically formulated as $\mathbf{H} = \text{softmax}(\mathbf{Z})$, $\mathbf{Z} = \sum_{l=1}^{L} \gamma_l \mathbf{H}^{(l)}$, and $\mathbf{H}^{(l)} = \sigma((\alpha_l \mathbf{PH}^{(l-1)} + (1 - \alpha_l)\mathbf{H}^{(0)})(\beta_l \mathbf{W}^{(l)} + (1 - \beta_l)\mathbf{I}))$, where $\alpha_l$, $\beta_l$ and $\gamma_l$ are the learnable parameters for the $l$-th layer of the model, and $\gamma_l \geq 0 \ \forall l \in \{0, 1, \ldots, L\}$, $\sum_{l=0}^{L} \gamma_l = 1$. Intuitively, $\alpha$ can determine the trade-off between topology information and the initial node information of each layer, which preserves node information while also expanding the receptive field. $\beta$ controls the model complexity, and $\gamma$ can aggregate each hidden layer with attention mechanism. In addition, GCNII has manually adjustable hyper-parameters $\alpha$ and $\beta$, DGCN with learnable parameters $\beta$ and $\gamma$, and ResGCN with the residual connections. Due to the space limit, the mathematical formulas and other details of the models are deferred to Appendix.

Table 1: Comparison of classification accuracy on citation network datasets.

| Dataset | Models | 2 Layers(epoch) | 4 Layers | 8 Layers | 16 Layers | 32 Layers | 64 Layers | 128 Layers(epoch) |
|---------|--------|-----------------|----------|----------|-----------|-----------|-----------|-------------------|
| Cora | GCN | 0.8010(600) | **0.8011(600)** | 0.7471(600) | FTT | – | – | – |
| | ResGCN | **0.8106(600)** | 0.7993(600) | 0.7735(600) | 0.7885(600) | 0.7793(1000) | FTT | – |
| | GCNII | 0.8006(200) | 0.7965(200) | 0.8004(200) | 0.8098(200) | 0.8219(200) | 0.8153(200) | **0.8233(200)** |
| | DGCN | 0.8094(200) | 0.8093(200) | 0.8113(200) | 0.8105(200) | 0.8062(200) | 0.8125(200) | **0.8132(200)** |
| | **ADGCN**$(\alpha, \beta, \gamma)$ | 0.7871(200) | 0.8147(200) | 0.8151(200) | 0.8152(200) | 0.8181(200) | 0.8214(200) | **0.8222(200)** |
| | ADGCN(uniform) | 0.7700(200) | 0.7993(200) | 0.7788(200) | 0.7898(200) | 0.7876(200) | 0.7987(200) | **0.8069(200)** |
| PubMed | GCN | **0.7301(600)** | 0.7292(600) | 0.6748(600) | FTT | – | – | – |
| | ResGCN | 0.7398(600) | **0.7678(600)** | 0.7590(600) | 0.7222(600) | 0.7275(600) | FTT | – |
| | GCNII | 0.7687(200) | 0.7699(200) | 0.7804(200) | 0.7962(200) | 0.7975(200) | **0.7984(200)** | 0.7949(200) |
| | DGCN | 0.7795(200) | 0.7725(200) | 0.7844(200) | 0.7838(200) | 0.7849(200) | 0.7854(200) | **0.7890(200)** |
| | **ADGCN**$(\alpha, \beta, \gamma)$ | 0.7653(200) | 0.7812(200) | 0.7902(200) | 0.7908(200) | 0.7922(200) | **0.7981(200)** | 0.7945(200) |
| | ADGCN(uniform) | 0.7668(200) | 0.7625(200) | 0.7706(200) | 0.7702(200) | 0.7731(200) | 0.7669(200) | **0.7736(200)** |

## 3 EXPERIMENT AND MAIN RESULT

We use the Cora, Citeseer, PubMed, and Wiki Chameleons datasets to assess the accuracy of each model in the node classification task. Due to space limitations, the statistical information of the datasets, detailed comparisons between different architectures, and the ablation experiments of the ADGCN model will be provided in the Appendix.

As shown in Table 1, we highlight important results in bold, and FTT stands for failure to train, indicating a model's inability to achieve satisfactory training accuracy within a reasonable number of iterations. In comparison to other models, ADGCN achieves higher accuracy with deeper layers

without having to manually adjust the hyper-parameters for every configuration. ADGCN performs comparably to GCNII and outperforms ResGCN and DGCN. The optimal accuracies for the GCN and ResGCN are between 2 and 8 layers, which are less than the optimal value for the GCNII, DGCN and ADGCN. The final row of each dataset indicates that the model's performance will be significantly influenced by the parameters' initialization method, which should be taken into consideration. We initialize $\alpha_l, \beta_l, \gamma_l$ to $\lambda(1 - \lambda)^l$, where $\lambda$ is a hyperparameter and the model is insensitive to it.

As observed from the degradation phenomenon of ResGCN, the addition of weight matrices results in excessive transformations, increasing model complexity and leading to overfitting issues. In contrast, the GCNII, DGCN, and ADGCN models, which possess the ability to adjust model complexity, demonstrate the capacity to mitigate model degradation. ADGCN specifically addresses the issue of information not smoothly propagating between layers in the neural network and the unique properties of graph-structured data, which includes both topology and node feature information. Through appropriate initialization settings, ADGCN achieves optimal performance. GCNII requires manual adjustment of layer-specific hyper-parameters and lacks information aggregation across layers, making it less suitable for processing large-scale graph-structured data. DGCN, due to its disregard for node information and inadequate parameter initialization, performs inferiorly compared to ADGCN in terms of performance. In general, the ADGCN model demonstrates high adaptability and scalability for handling large-scale data while effectively overcoming model degradation problem.

## URM STATEMENT

Author Sisi Zhang meets the URM criteria of the ICLR 2023 Tiny Papers Track.

## ACKNOWLEDGEMENTS

This work is funded by the National Natural Science Foundation of China (62272093, 62137001).

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

## A    APPENDIX

### A.1    GCN AND ITS VARIANTS

The following architectures are the GCN and its variants.

**GCN.**    Kipf & Welling (2017) indicated that the GCN is the one-order approximation of spectral GCN with propagation operation $\mathbf{P}$ and transformation operation $\mathbf{W}$. The normalized propagation matrix (i.e., graph convolution operation) $\mathbf{P} = \mathbf{I_N} + \mathbf{D}^{-\frac{1}{2}}\mathbf{A}\mathbf{D}^{-\frac{1}{2}}$, where $\mathbf{A}$ denotes adjacency matrix and $\mathbf{D}$ denotes the diagonal degree matrix. The output of $l$-th graph convolution layer is formulated as the following:

$$\mathbf{H}^{(l)} = \sigma(\mathbf{P}\mathbf{H}^{(l-1)}\mathbf{W}^{(l)}), \tag{1}$$

where $\sigma(\cdot)$ is ReLU activation, and $\mathbf{W}^{(l)}$ is the weight matrix of $l$-th layer.

**ResGCN.**    Li et al. (2019) suggest GCN combined with residual connections to address the the gradient vanishing during deepening. The output of $l$-th layer is formulated as the following:

$$\mathbf{H}^{(l)} = \sigma(\mathbf{P}\mathbf{H}^{(l-1)}\mathbf{W}^{(l)}) + \mathbf{H}^{(l-1)}. \tag{2}$$

The ResGCN model can solve the gradient vanishing, but we still observe model degradation issue in Table 1 and Table 3.

**GCNII.**    Chen et al. (2020b) is one of the state-of-the-art deep GCNs so far, which is composed of initial residual and identity mapping techniques. To achieve optimal generalization performance, the critical parameters (i.e., $\alpha$ and $\beta$) for different datasets must be manually adjusted. The $l$-th layer is formulated as the following:

$$\mathbf{H}^{(l)} = \sigma\Big(\big((1-\alpha_l)\mathbf{P}\mathbf{H}^{(l-1)} + \alpha_l\mathbf{H}^{(0)}\big)\big((1-\beta_l)\mathbf{I}_n + \beta_l\mathbf{W}^{(l)}\big)\Big), \tag{3}$$

where $\alpha_l \in [0, 1]$ and $\beta_l \in [0, 1]$.

**DGCN.** Cong et al. (2021) computes the output of the model with learnable parameters $\alpha_l$ and $\beta_l$. By separating expressive power from generalization ability, it creates a decoupled structure for GCNs. This can lessen the generalization gap's reliance on the weight matrices. It is defined as:

$$\mathbf{Z} = \sum_{l=1}^{L} \alpha_l \mathbf{H}^{(l)}, \ \mathbf{H}^{(l)} = \mathbf{P}^l \mathbf{H}^{(l-1)}((1-\beta_l)\mathbf{I}_n + \beta_l \mathbf{W}^{(l)}), \tag{4}$$

where $\beta_l \in [0, 1]$.

## A.2 ILLUSTRATING THE MODEL DEGRADATION

### A.2.1 EXPERIMENT SETUP

Here are the models' additional empirical evaluations using the widely used standard citation network datasets and Wiki Chameleons dataset. Statistics of the datasets are summarized in Table 2 (Sen et al., 2008; Rozemberczki et al., 2021). We consider semi-supervised node classification tasks and contrast how the models' generalization performance changes as the number of layers increases. We set the hidden dimension as 64, dropout rate as 0.1, learning rate as 0.01, $\lambda = 0.1$.

Table 2: Dataset statistics.

| Dataset | Type | Nodes | Edge | Classes | Features |
|---|---|---|---|---|---|
| Cora | Citation network | 2,708 | 10,556 | 7 | 1,433 |
| Citeseer | Citation network | 3,327 | 9,104 | 6 | 3,703 |
| PubMed | Citation network | 19,717 | 88,648 | 3 | 500 |
| Wiki Chameleons | Knowledge graph | 2,277 | 36,101 | 5 | 2,325 |

### A.2.2 RESULTS

Table 3 is the supplementary results of Table 1 in section 3. The experimental results of various models on the Citeseer and Wiki datasets are consistent with those on the Cora and PubMed datasets, with the ADGCN model performing the best. It is evident that all improved models show significantly higher accuracy compared to the GCN model. It should be noted that the Citeseer dataset is denser than Cora and PubMed, resulting in more pronounced improvements for various models on it, which is in line with intuition. Additionally, the GCNII model did not undergo precise manual parameter tuning for the Wiki dataset, resulting in relatively poorer performance, highlighting the high adaptability of the ADGCN model.

We conduct ablation experiments on the adaptive parameters of the ADGCN model and analyzed their respective effects. As shown in Table 4, overall, the ADGCN model demonstrates more stable performance compared to the various settings in the ablation experiments, indicating the importance of each component of the ADGCN model. Specifically, the experimental results with only the $\alpha$, $\beta$, and $\gamma$ parameters retain exhibit model degradation problem. The ADGCN model with the retention of $\beta$ and $\gamma$ parameters successfully overcomes the degradation issue. The addition of the $\alpha$ parameter achieves the maximum improvement in model performance. This indicates a progressive relationship between the adaptive parameters. By selectively aggregating inter-layer feature information using the $\gamma$ parameter to ensure smooth model training, overcoming the degradation issue of deep models is further achieved with the $\beta$ parameter, and extracting as many features as possible from the graph-structured data is accomplished with the $\alpha$ parameter. This approach leads to the attainment of a model with maximum potential.

Figures 1 to 4 provide a detailed depiction of the generalization performance variation as the number of layers increases from 2 to 48, showcasing the fine-grained changes in node classification performance for different architectures. They compare the results across different datasets and demonstrate similar outcomes. These findings confirm the effectiveness and robustness of the ADGCN model. They also highlight the importance of the $\alpha$, $\beta$, and $\gamma$ parameters in the ADGCN, as they play significant roles in capturing graph information, exploring model depth, and facilitating effective information propagation between layers.

Table 3: Accuracy for the node classification task.

| Datasets | Models | 2 Layers(epoch) | 4 Layers | 8 Layers | 16 Layers | 32 Layers | 64 Layers | 128 Layers(epoch) |
|---|---|---|---|---|---|---|---|---|
| | GCN | **0.6298(600)** | 0.6010(600) | 0.5745(600) | FTT | – | – | – |
| | ResGCN | 0.6927(600) | 0.6801(600) | **0.6994(600)** | 0.6889(600) | 0.6353(1000) | FTT | – |
| Citeseer | GCNII | 0.6843(200) | 0.6687(200) | 0.6864(200) | 0.6838(200) | 0.6949(200) | 0.6955(200) | **0.6971(200)** |
| | DGCN | 0.6935(200) | 0.6917(200) | 0.6894(200) | 0.6919(200) | 0.6891(200) | 0.6953(200) | **0.6980(200)** |
| | **ADGCN**$(\alpha,\beta,\gamma)$ | 0.6933(200) | 0.6951(200) | 0.7082(200) | 0.7002(200) | 0.7030(200) | 0.7031(200) | **0.7118(200)** |
| | ADGCN(uniform) | 0.6433(200) | 0.6630(200) | 0.6605(200) | 0.6845(200) | **0.7081(200)** | 0.6917(200) | 0.6746(200) |
| | GCN | 0.3822(1000) | 0.3879(1000) | 0.4279(1000) | FTT | – | – | – |
| | ResGCN | 0.3750(1000) | 0.3934(1000) | 0.3810(1000) | 0.3843(1000) | 0.3541(1000) | FTT | – |
| Wiki | GCNII | 0.3867(400) | 0.3923(400) | 0.4228(1000) | 0.3973(1000) | 0.4067(1000) | 0.4072(1000) | 0.4158(1000) |
| | DGCN | 0.3918(400) | 0.3663(400) | 0.3509(1000) | FTT | – | – | – |
| | **ADGCN**$(\alpha,\beta,\gamma)$ | 0.4741(400) | 0.4638(400) | 0.4712(1000) | 0.4743(1000) | **0.4788(1000)** | 0.4689(1000) | 0.4648(1000) |
| | ADGCN(uniform) | 0.4386(400) | 0.4796(400) | 0.4419(400) | 0.4657(400) | 0.4718(400) | 0.4433(400) | 0.4567(400) |

Table 4: Ablation experiments.

| Datasets | Models | 2 Layers(epoch) | 4 Layers | 8 Layers | 16 Layers | 32 Layers | 64 Layers | 128 Layers(epoch) |
|---|---|---|---|---|---|---|---|---|
| | GCN | 0.8010(600) | **0.8011(600)** | 0.7471(600) | FTT | – | – | – |
| | ADGCN$(\alpha)$ | **0.7949(200)** | 0.7871(200) | 0.7214(200) | 0.7662(200) | 0.7376(200) | 0.7495(200) | 0.7733(200) |
| | ADGCN$(\beta)$ | **0.8115(200)** | 0.8014(200) | 0.7942(200) | 0.7954(200) | 0.6833(200) | FTT(200) | FTT(200) |
| Cora | ADGCN$(\gamma)$ | **0.7978(200)** | 0.7893(200) | 0.7818(200) | 0.7841(200) | 0.7917(200) | 0.7803(200) | 0.7913(200) |
| | ADGCN$(\alpha,\beta)$ | 0.7840(100) | 0.8077(100) | 0.8192(100) | **0.8215(100)** | 0.8185(100) | 0.8137(100) | 0.8137(100) |
| | ADGCN$(\alpha,\gamma)$ | 0.7987(200) | **0.8003(200)** | 0.7787(200) | 0.7542(200) | 0.7575(200) | 0.7903(200) | 0.7871(200) |
| | ADGCN$(\beta,\gamma)$ | 0.8094(200) | 0.8093(200) | 0.8113(200) | 0.8105(200) | 0.8062(200) | 0.8125(200) | **0.8132(200)** |
| | **ADGCN**$(\alpha,\beta,\gamma)$ | 0.7871(100) | 0.8147(100) | 0.8151(200) | 0.8152(200) | 0.8181(200) | 0.8214(200) | **0.8222(200)** |
| | GCN | **0.6298(600)** | 0.6010(600) | 0.5745(600) | FTT | – | – | – |
| | ADGCN$(\alpha)$ | 0.6661(200) | **0.6880(200)** | 0.6628(200) | 0.6529(200) | 0.6473(200) | 0.6704(200) | 0.6607(200) |
| | ADGCN$(\beta)$ | **0.6911(200)** | 0.6867(200) | 0.6771(200) | 0.6576(200) | 0.4909(200) | FTT(200) | FTT(200) |
| Citeseer | ADGCN$(\gamma)$ | 0.6695(200) | 0.6477(200) | 0.6696(200) | **0.6742(200)** | 0.6607(200) | 0.6651(200) | 0.6712(200) |
| | ADGCN$(\alpha,\beta)$ | 0.6925(200) | 0.6897(200) | 0.6943(200) | 0.6971(200) | 0.6924(200) | **0.6983(200)** | 0.6942(200) |
| | ADGCN$(\alpha,\gamma)$ | 0.6763(200) | 0.6847(200) | **0.6957(200)** | 0.6856(200) | 0.6508(200) | 0.6714(200) | 0.6763(200) |
| | ADGCN$(\beta,\gamma)$ | 0.6935(200) | 0.6917(200) | 0.6894(200) | 0.6919(200) | 0.6891(200) | 0.6953(200) | **0.6980(200)** |
| | **ADGCN**$(\alpha,\beta,\gamma)$ | 0.6933(200) | 0.6951(200) | 0.7082(200) | 0.7002(200) | 0.7030(200) | 0.7031(200) | **0.7118(200)** |
| | GCN | **0.7301(600)** | 0.7292(600) | 0.6748(600) | FTT | – | – | – |
| | ADGCN$(\alpha)$ | 0.7612(200) | **0.7648(200)** | 0.7305(200) | 0.7341(200) | 0.7516(200) | 0.7454(200) | 0.7484(200) |
| | ADGCN$(\beta)$ | 0.7743(200) | 0.7704(200) | 0.7779(200) | **0.7833(200)** | 0.7735(200) | FTT | – |
| PubMed | ADGCN$(\gamma)$ | **0.7817(200)** | 0.7641(200) | 0.7794(200) | 0.7734(200) | 0.7637(200) | 0.7633(200) | 0.7544(200) |
| | ADGCN$(\alpha,\beta)$ | 0.7557(200) | 0.7878(200) | 0.7898(200) | **0.7939(200)** | 0.7906(200) | 0.7921(200) | 0.7838(200) |
| | ADGCN$(\alpha,\gamma)$ | 0.7734(200) | 0.7610(200) | **0.7746(200)** | 0.7677(200) | 0.7648(200) | 0.7498(200) | 0.7558(200) |
| | ADGCN$(\beta,\gamma)$ | 0.7795(200) | 0.7725(200) | 0.7844(200) | 0.7838(200) | 0.7849(200) | 0.7854(200) | **0.7890(200)** |
| | **ADGCN**$(\alpha,\beta,\gamma)$ | 0.7653(200) | 0.7812(200) | 0.7902(200) | 0.7908(200) | 0.7922(200) | **0.7981(200)** | 0.7945(200) |

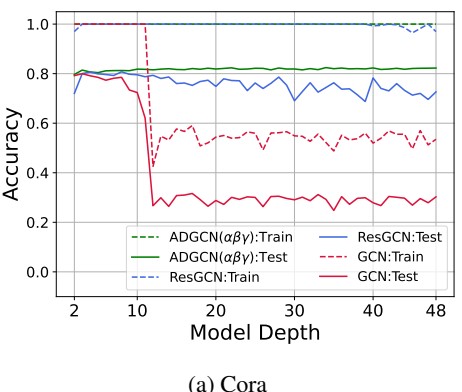

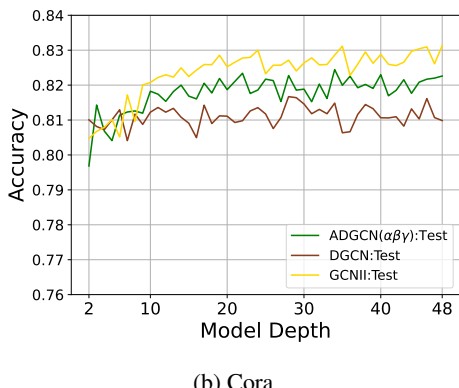

(a) Cora

(b) Cora

Figure 1: On the Cora dataset, (a) compare the ADGCNs with learnable parameters, ResGCNs with residual connections and standard GCNs respectively. Note that the model degradation is reflected in the red solid line in ranges 2 to 11 and the blue solid line in ranges 2 to 48 showing a slightly decline in testing accuracy, meanwhile, the training accuracy roughly equal to 1. (b) compare the ADGCNs, DGCNs and GCNIIs in testing accuracy in ranges 2 to 48. The performance of ADGCN is superior to that of DGCN and nearly equal to that of GCNII with manual parameter adjustment. There is no model degradation in any of the three models.

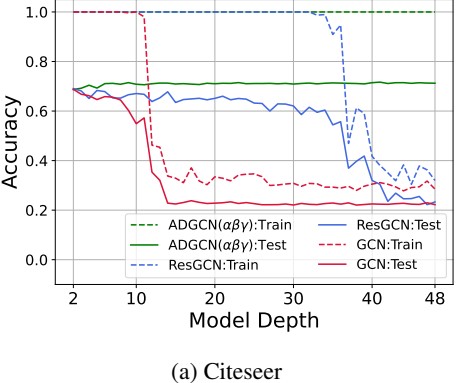

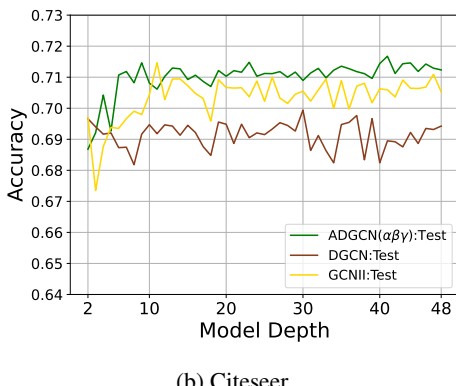

(a) Citeseer

(b) Citeseer

Figure 2: On the Citeseer dataset, (a) compare the ADGCNs, ResGCNs and standard GCNs respectively. Note that the experimental results are similar to Cora. (b) compare the ADGCNs, DGCNs and GCNIIs in testing accuracy in ranges 2 to 48. The performance of ADGCN is superior to that of DGCN and that of GCNII with manual parameter adjustment. There is no model degradation in any of the three models, meanwhile, ADGCN and GCNII are more stable than DGCN.

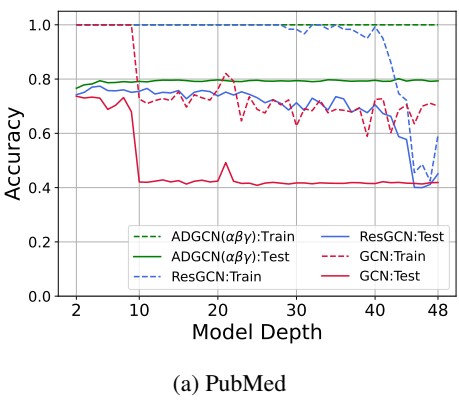

(a) PubMed

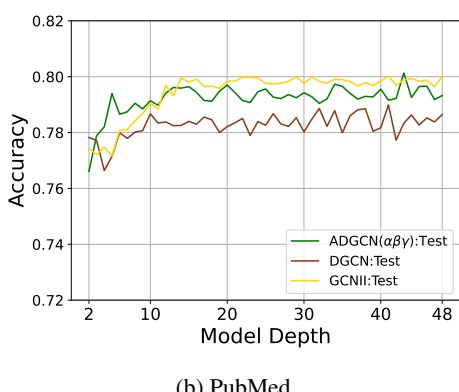

(b) PubMed

Figure 3: On the PubMed dataset, (a) compare the ADGCNs, ResGCNs and standard GCNs respectively. Note that the experimental results are similar to Cora and Citeseer. (b) compare the ADGCNs, DGCNs and GCNIIs in testing accuracy. The performance of ADGCN is superior to that of DGCN and nearly equal to that of GCNII. There is no model degradation in any of the three models.

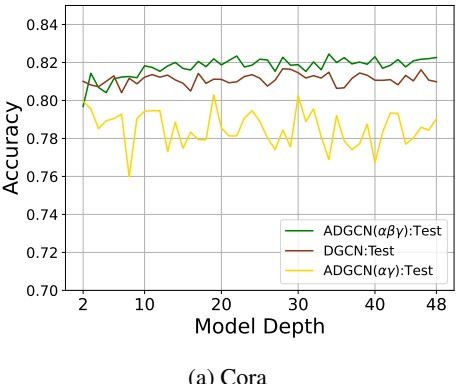

(a) Cora

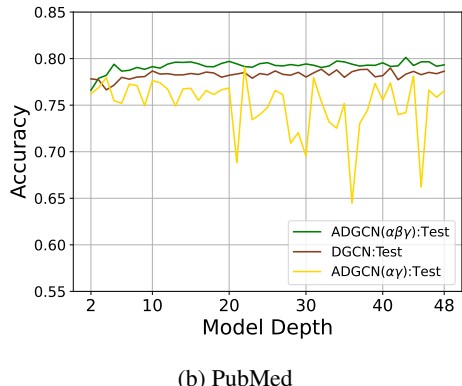

(b) PubMed

Figure 4: Comparison of models with different parameter configurations. The yellow curves show instability compared with the others . This demonstrates the necessity of controlling model complexity through $\beta$.

