# OpenReview forum: "Propagate Deeper and Adaptive Graph Convolutional Networks"
_ICLR.cc/2023/TinyPapers — Submitted to Tiny Papers @ ICLR 2023_

### Official Review · Reviewer_nghd · 2023-03-29

**Confidence:** 4

**Summary Of Contributions:**

This paper attempt to addresses the problem of performance degradation in GCNs as the number of layers increases. The proposed method introduce three adaptive parameters to realize a Deep GCN architecture.

**Rating:**

Clear, Correct, and Reproducible (CCR): a submission which meets the reviewing criteria

**Strengths And Weaknesses:**

[Advantages]:
- This paper is well-written and easy to follow.
- The experimental results demonstrate the potential of the proposed method.


[Disadvantages]:
- Overall, the work simply combines GCNII and DGCN. The author should provide more insights on the shortcomings of previous methods.
- The three adaptive parameters are not orthogonal, where the author should explain more about their relationship. Which one alleviate the over-smoothing problem? Which one alleviate the gradient vanishing problem?

**Suggested Changes:**

- The author should conduct ablation experiments for each adaptive parameter and analyze their respective effects.

---

### Official Review · Reviewer_Y5EG · 2023-03-30

**Confidence:** 4

**Summary Of Contributions:**

This paper proposes an adaptive deep convolutional network (ADGCN) architecture to alleviate the performance degradation of graph convolutional networks (GCNs) as the number of layers increases. ADGCNs can be adaptively tuned to different graph structure data. The experiments are conducted on some real datasets.

**Rating:**

Clear, Correct, and Reproducible (CCR): a submission which meets the reviewing criteria

**Strengths And Weaknesses:**

Strength:

•	The idea of eliminating the equivalence between model degradation and model over-smoothing is interesting and promising.

•	The submission follows the basic requirements of format and page count. Follow the ICLR code of conduct.

Weakness:

•	Although many experimental results are provided in the paper, the experimental results need to be sharper to highlight the effect.


**Suggested Changes:**

•	The motivation and research questions of this paper are strong and important. My main concern is that the technical contribution may not be significant.

•	Comparative experiments of the framework proposed in the paper and the effect of the architecture modified with manual parameters can be added.

---

### Comment · Area_Chair_FD9f · 2023-06-06
**Archival**

 This work meets the threshold for archival, contains the URM statement and is deanonymized.

---

### Meta-Review · Area_Chair_FD9f · 2023-04-05

**Recommendation:** Invite to present
**Confidence:** 4

**Metareview:**

- Clarity: The paper is clear: the problem is well introduced, and the methodology is explained in detail. The literature discussion might be slightly sharper in commenting on different prior works, but it is acceptable given the paper space.
- Correctness: The paper seems correct, and the experiments are pretty extensive to support the claims; further discussion and analysis on parameters combination could be beneficial.
- Reproducibility: The paper contains enough details to reproduce a similar experimental setting, but exact replication is not possible in the current status. The code is not mentioned in the manuscript.

The reviewers appreciated the research question, the clarity, and part of the experiments, while further analysis would help to shape the final paper message.

**Summary:**

The paper proposes an analysis of different Graph Convolutional Networks, analyzing the impact of three different parameters. The paper is clear, and the research question is interesting, but further investigation on different parameter tuning could improve the takeaway message.

**Comments And Feedback To The Authors:**

The general feeling on the paper is positive, but the technical contribution is still not completely clear. I suggest complementing the analysis with the reviewers' comments. Especially further analysing the parameters' roles and their values would be an interesting ablation. The paper closes with a quite general message: "A stable and adaptable architecture can be obtained on the assumption that each component
is given careful consideration"; it would be more interesting if something could be said in concrete. What is the main promising direction highlighted by the paper? What should future works investigate?

I also suggest elaborating a bit more on the previous literature and methods, providing a more detailed comment for the different works and highlighting their limitations that this work aims to solve.

Minor:
- Typo: Introdution -> Introduction

**Reason For Not Giving A Higher Recommendation:**

A clear final message of the paper is missing, and reviewers have concerns about the actual technical contribution. Further experimentations with the parameters with a more clear scope might help to sharpen the paper takeaway.

**Reason For Not Giving A Lower Recommendation:**

The paper reaches the bar of Clear, Correctness, and Reproducibility, it is well-presented, and the experimentations (while preliminary) are satisfactory.

---

### Decision · Program_Chairs · 2023-04-08

Invite to present